# Cyclic loading test study on a new cast-in-situ insulated sandwich concrete wall

**Wentao Qiao**[1,2]*, **Xiaoxiang Yin**[1], **Shengying Zhao**[3], **Dong Wang**[4]

**1** School of Civil Engineering, Shi Jiazhuang Tiedao University, Shi Jiazhuang, China, **2** Cooperative Innovation Center of Disaster Prevention and Mitigation for Large Infrastructure in Hebei Province (Shi Jiazhuang Tiedao University), Shi Jiazhuang, China, **3** School of Civil Engineering, Harbin Institute of Technology, Harbin, China, **4** TRC Companies, Baton Rouge, United States of America

* tottyer@126.com

**Data Availability Statement:** All relevant data are within the paper and its Supporting Information files.

**Funding:** This research was funded by Natural Science Foundation of HeBei Province

## Abstract

Insulated sandwich concrete panel (ISCP) is widely used because of its high thermal insulation efficiency and low construction cost. Aiming at improving traditional ISCP, a new cast-in-situ concrete wall structure made of ISCP is proposed, which is composed of thin-walled cold-formed steels, slant steel wire connectors, steel wire meshes, concrete layers, expanded polystyrene sheets and reinforced concrete embedded columns. In order to assess the hysteretic properties of the new insulated sandwich concrete wall and the influence of various parameters, low-frequency horizontal cyclic load tests were carried out on seven full-scale specimens of new type cast-in-situ insulated sandwich concrete wall. The specimens were compared and analyzed with respect to failure mode, bearing capacity, ductility, degradation characteristics and energy dissipation capacity. The results show that the final failure pattern of the specimen is two main diagonal cracks intersecting each other; the bearing capacity is greatly affected by concrete thickness and axial compression ratio, regardless of concrete strength. Brittle failure is typically observed when the steel wire spacing is large, while ductility is pronounced when the concrete layer thickness is small and the concrete strength is low; the smaller the thickness of concrete layer, the faster the stiffness degrades. The wall structure shows a better energy dissipation performance with a smaller steel wire spacing, lower concrete strength and smaller axial compression ratio.

## Introduction

Nowadays, environmental protection and sustainable development are hot issues of global concern. In the architectural field, more and more scholars focus on the study of energy-saving buildings. Insulated sandwich concrete panel (ISCP) is representative of an energy-saving building, which is typically composed of concrete layers on both sides, insulation layers in between, and shear connectors. The main advantages of this panel are: (i) ease of erection at reasonable costs, (ii) high efficiency of thermal and sound insulation, and (iii) flexibility in utility, either as enveloping elements or as load-bearing components.

(E2016210052), Science and Technology Research Key Project in Higher Institutions of Hebei Province (ZD2018250). Additionally, this research was also conducted with the financial support of Hebei Xi Jiefa Construction Engineering Co., Ltd. The TRC Companies provided support in the form of salaries for author D.W. The funders had no role in study design, data collection and analysis, decision to publish, or preparation of the manuscript. The specific roles of these authors are articulated in the 'author contributions' section.

**Competing interests:** This research was also conducted with the financial support of Hebei Xi Jiefa Construction Engineering Co., Ltd. The TRC Companies provided support in the form of salaries for author D.W. This does not alter our adherence to PLOS ONE policies on sharing data and materials.

To date, research on ISCP mainly focuses on efficiency of thermal insulation, composite action, materials and constructions and structural performance.

In terms of insulation efficiency, how to reduce the influence of cold bridge effect is mainly considered. Currently, the most popular method is to make shear connectors with materials with low thermal efficiency. For instance, a new type of hybrid connector wrapped with nylon was proposed to improve the thermal performance of ISCP wall [1]. Other commonly used materials with low thermal efficiency include fiber-reinforced polymer (FRP), carbon-fiber-reinforced polymer (CFRP), and glass-fiber-reinforced polymer (GFRP).

In terms of composite action, ISCP are usually categorized into non-composite (NC), partially composite (PC) and fully composite (FC) according to the degree of composite action (i.e., the amount of longitudinal shear transfer between concrete layers through the insulation layer). It is generally believed that the composite action of ISCP is mainly provided via shear connectors, and studies indicate that connectors' dimension and spacing greatly affect composite action, and that connectors inserted at an angle other than 90° are considerably stronger than those inserted perpendicularly [2]. Most studies on composite action are considered from the flexural response and shear response [3–6].

In terms of materials and constructions, many researchers proposed different forms of materials and constructions. The changes of materials and constructions for better mechanical performance, higher insulation efficiency and easier construction, but they often restrict each other in practice. In order to make the construction operation easier, the sandwich concrete panel without shear connector was proposed [7,8], and textile reinforced concrete (TRC) layer was adopted, but its bearing capacity was too low. In order to improve bearing capacity, Xie Qun et al. adopted the reinforced concrete core column, and bearing capacity has been effectively improved [9]. Thomas G. Norris et al. tried to improve the performance of sandwich concrete panel by using the function of restrained concrete, but its construction form was too complex, which was inconvenient for construction [10]. In addition, some researchers have focused on shear connectors to improve the performance of sandwich concrete panels by changing the materials of the connectors [11–13], shape and layout [14–16]. These structures are very novel, but they are very complicated to manufacture and cost a lot. Some other studies focus on non-load-bearing components and use textile reinforced reactive powder concrete (TRRPC) as the vertical panel and GFRP as the connectors, which improves the cracking and insulation performance of the sandwich concrete panel [17,18]. A lot of bending, shearing and axial compression tests have been carried out in the above aspects of research. However, there are few studies on the seismic behavior of ISCP. Studies have shown that the seismic performance of ISCP is comparable with those of common reinforced concrete panels [19,20]. Moreover, according to the results of the shaking table test, the stiffness and strength of ISCP structure under the action of dynamic excitation are better than the previous results of the quasi-static cycle test of a single specimen [21]. In addition, ISCP also shows good performance when it is used for filling walls [22] and prefabricated components [23]. Even under the explosion load, ICSP wall shows a stronger ability to absorb and dissipate the explosion energy and has a better anti-crushing stability [24].

Existing insulated sandwich concrete structure systems often lead to the cold bridge effect due to the discontinuity of the insulation layer at the joints of walls, and most of the joints are complex in composition and inconvenient in construction. Although the thermal efficiency problem can be solved by using FRP connectors, FRP connectors are prone to brittle fracture and be pulled-out, and the cost is high [3,12]. In order to solve the problems above, this paper proposes a new type of cast-in-situ insulated sandwich concrete wall structure by adding thin-walled cold-formed steels, reinforced concrete embedded columns and changing the layout of connectors based on traditional ISCP. The structure is composed of thin-walled cold-formed

steels, slant steel wire connectors, steel wire meshes, concrete layers, expanded polystyrene sheets and reinforced concrete embedded columns, which has been significantly improved in terms of thermal insulation efficiency, structural optimization and cost control. Main features of this novel structure are:

- All the walls are sandwich walls, composed of three-dimensional steel wire skeletons, expanded polystyrene sheets, concrete and thin-walled cold-formed steels.

- The two concrete layers of the wall are connected by slant steel wires used as shear connectors. The angle between the slant steel wire and the wall is 72˚, and the layout spacing is 50 mm. The wall belongs to partially composite sandwich structure.

- The thickness of concrete layers on both sides is identical, as is the ratio of horizontal and vertical reinforcement.

- In the openings, centers and edges of the sandwich walls, closed frames are formed by horizontally and vertically welding thin-walled cold-formed steels with rectangular sections.

- The walls are connected by reinforced concrete embedded columns and additional reinforcements.

Seven full-scale specimens of the wall were made and tested under horizontal cyclic load in-plane, to investigate the potential effect of four variables (thickness of concrete layers, spacing of steel wires, concrete strength and axial compression ratio), and to evaluate the seismic performance of the new structure.

The remainder of this paper is organized as follows: Section 2 introduces the construction of this new-style structure, Section 3 describes the key points of the cyclic loading experiment, and Section 4 presents the failure patterns and mechanisms. In Section 5, the hysteretic curve, load-bearing capacity, ductility, stiffness degradation and energy dissipating capacity are analyzed based on the experimental results. The anti-seismic performance and influence of the main parameters are discussed in Section 6.

## The structure system

### Cast-in-situ insulated sandwich concrete walls

A cast-in-situ insulated sandwich concrete wall is composed of an expanded polystyrene sheet, steel wires, rectangular steel tubes, additional reinforcements and concrete layers, as shown in Fig 1.

A polystyrene sheet used as an insulation layer has a thickness of 80 mm and density of 20.4 kg/m$^3$.

The steel wires are made of low carbon galvanized steel wires, including horizontal steel wires, vertical steel wires and slant steel wires, usually with a diameter of 2.5 mm. The horizontal steel wires and vertical steel wires are spot welded together at the intersection, with a steel wire spacing of 50 mm. The slant steel wires travel through the polystyrene sheet and the two ends are spot welded on the steel wire meshes composed of horizontal steel wires and vertical steel wires on both sides of the wall; the adjacent two columns of slant steel wires have opposite directions, and the angle between the slant steel wire and the polystyrene sheet is 72˚, the spacing of slant steel wires is 50 mm, as shown in Fig 1. These operations are done mechanically, resulting in a polystyrene sheet welded to the steel wire meshes at the designed position.

Thin-walled cold-formed steel tubes are adopted with rectangular sections (70 mm×40 mm×1.5 mm), which are arranged in the openings of doors and windows, the center and

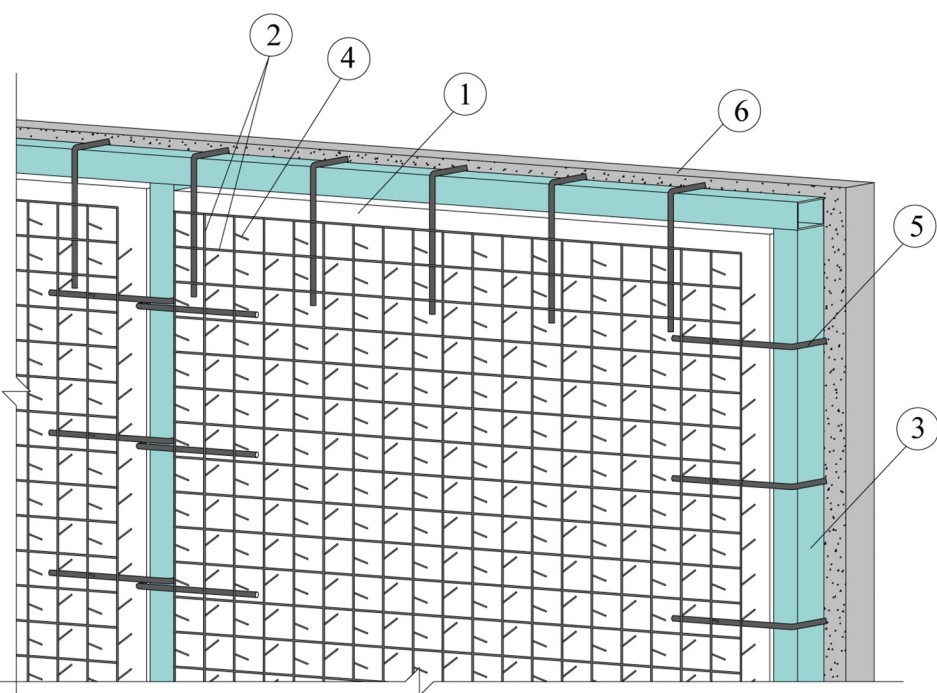

① Expanded polystyrene sheet ② Galvanized welded steel wire meshes
③ Thin-walled cold-formed steel tubes with rectangular sections
④ Slant steel wire connectors ⑤ Additional reinforcements
⑥ Concrete layers

**Fig 1. Layout of cast-in-situ insulated sandwich concrete walls.**

edges of the wall, and welded on the steel tubes additional reinforcements that extend into the wall. Additional reinforcements are plain bars with a diameter of 6 mm and a spacing of 300 mm, as shown in Fig 1.

The thickness of concrete layers on both sides is identical (50 mm), as shown in Fig 1. Pea gravel concrete of C30 and below (according to GB/T 50081–2002 Chinese standard for test method of mechanical properties on ordinary concrete) is used, and the maximum particle size of coarse aggregate is limited to 10 mm. Concrete is constructed in situ.

## Connecting joints of walls

The walls mainly bear vertical load and horizontal load. In order to ensure the complete transfer of horizontal load at the wall joint, the joint may bear bending, shearing and axial force. Therefore, different connection patterns are proposed for different joint forms. Four types of joint are used in this study: continuous joint of two walls ('–' shape), orthogonal joints of two walls ('L' shape), connecting joints of three walls ('T' shape) and orthogonal joints of four walls ('+' shape). Each type of joint targets at an overall performance regarding mechanical behavior of structure, thermal efficiency, feasibility of construction and economic feature.

- The continuous joints of two walls ('–' shape) are constructed by setting formed steel and additional reinforcements at the joint, as shown in Fig 2(a).

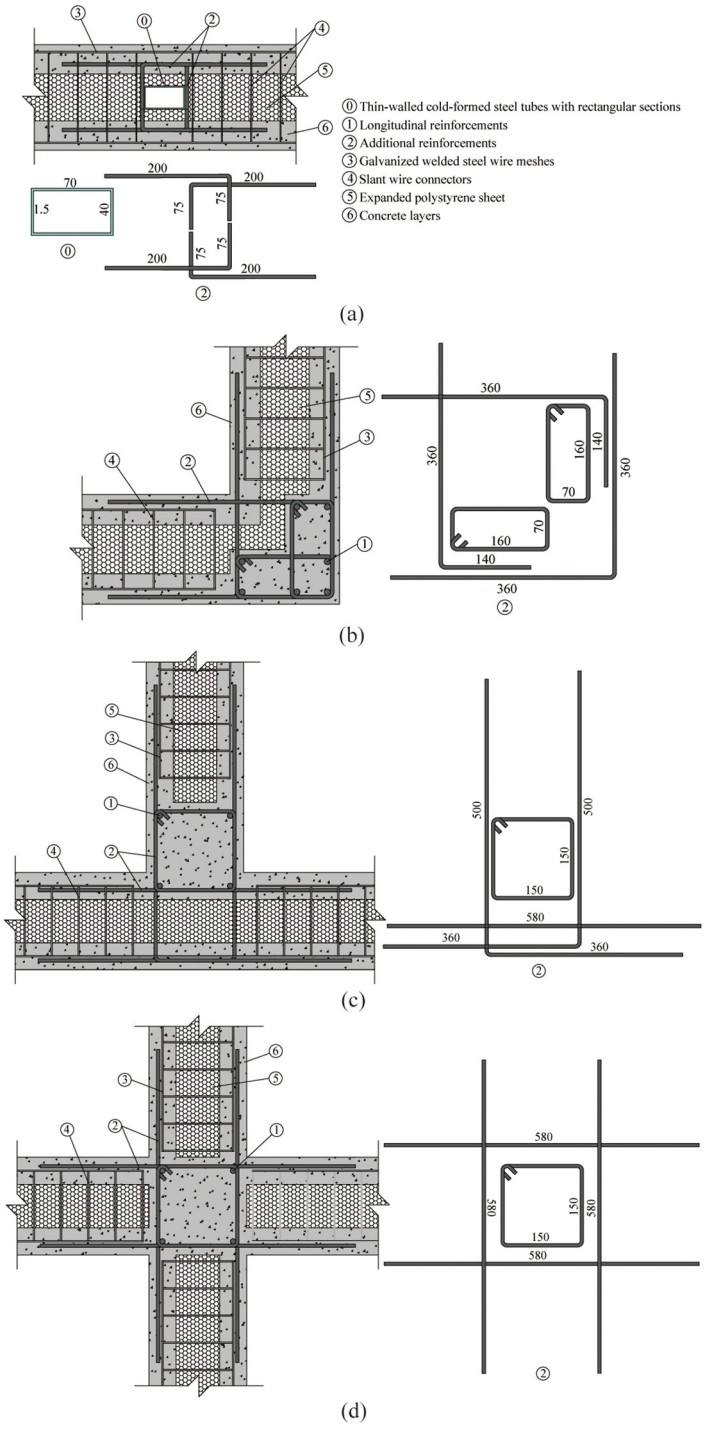

**Fig 2. Connection joints of walls (unit: mm).** (a) the continuous joint of two walls; (b) the orthogonal joints of two walls; (c) the connecting joints of three walls; (d) orthogonal joints of four walls.

- The orthogonal joints of two walls ('L' shape) are constructed by setting reinforced concrete special-shaped columns and additional reinforcements at the junction of the two walls, as shown in Fig 2(b).

- The connecting joints of three walls ('T' shape) and orthogonal joints of four walls ('+' shape) are constructed by setting reinforced concrete columns with additional reinforcements extended into each wall at the joint, as shown in Fig 2(c) and 2(d).

## Experimental plan

The main purpose of the experiment is to observe the pseudo-static behavior of the single cast-in-situ insulated sandwich concrete walls under low-frequency horizontal cyclic load, to reveal how various parameters influence the mechanical properties and deformation of the wall, and to evaluate the seismic performance of the structure. With this aim, a full-scale test of quasi-static force is carried out.

### Design of the experiment

Four parameters were varied: thickness of concrete layer $t$, steel wire spacing $s$, concrete strength $f_c$ and axial compression ratio $\lambda$. Accordingly, a total of seven specimens were made for experimental research. Parameters of each specimen is shown in Table 1.

Low frequency horizontal cyclic load and constant vertical load are applied to each specimen. The horizontal cyclic load is controlled by displacement, and the displacement of each level circulates twice until the horizontal load value drops to 85% of the peak load, so as to observe the pattern of strength degradation, stiffness degradation and damage development induced by repeated loading and unloading. The magnitude of vertical load is determined by the axial compression ratio. Two axial compression ratios are adopted, respectively representing different upper wall loads, to assess the influence of axial force.

Throughout the test, the relationship curve between load and displacement and failure mode are documented, which will be used to evaluate the mechanical properties of the wall and the influence of various research parameters.

### Fabrication of specimens

All seven specimens measure 1900 mm in height and 1000 mm in width. All specimens have a total wall thickness of 230 mm (75+80+75) except for CWS-2 (210 = 65+85+65 mm) and

**Table 1. Design details of the specimens.**

| Specimens | Wall dimensions | | | Concrete thickness $t$/mm | Steel wires spacing $s$/mm | Concrete strength $f_c$/MPa | Axial compression ratio /$\lambda$ |
|---|---|---|---|---|---|---|---|
| | $d$/mm [a] | $h$/mm [b] | $h_e$/mm [c] | | | | |
| CWS-1 | 230 | 1900 | 1500 | 75 | 50 | C30 | 0.06 |
| CWS-2 | 210 | 1900 | 1500 | 65 | 50 | C30 | 0.06 |
| CWS-3 | 250 | 1900 | 1500 | 85 | 50 | C30 | 0.06 |
| CWS-4 | 230 | 1900 | 1500 | 75 | 100 | C30 | 0.06 |
| CWS-5 | 230 | 1900 | 1500 | 75 | 50 | C20 | 0.06 |
| CWS-6 | 230 | 1900 | 1500 | 75 | 50 | C40 | 0.06 |
| CWS-7 | 230 | 1900 | 1500 | 75 | 50 | C30 | 0.12 |

[a] Total wall thickness, the sum of insulation layer thickness and concrete thickness on both sides.

[b] Total height of the wall, including a fixed region of 300 mm at the top, 100 mm at the bottom and 1500 mm at the center.

[c] Height of test region.

CWS-3 (250 = 85+80+85 mm), as shown in Table 1. The specimens are divided into 3 parts vertically: loaded region (top 300 mm), supported region (bottom 100 mm), and tested region (1500 mm in between).

The specimens are fabricated in the following procedures.

- Firstly, expanded polystyrene sheet (EPS) with steel wire arranged according to design is obtained through automatic processing machine: the spacing between horizontal steel wire and vertical steel wire is 50 mm, the horizontal and vertical spacing of the slant steel wire is also 50 mm, the angle between the slant steel wire and EPS plate is 72˚. The thickness of expanded polystyrene is 80 mm, as shown in Fig 3(a). The EPS and steel wires are provided by Hebei Meizhu Energy Saving Technology Co. LTD.

- Then, the reinforcements of the foundation beam and additional reinforcements are installed and tied. As shown in Fig 4 for detailed reinforcements of all specimens: at the junction between the wall and the foundation beam, additional reinforcements with a diameter of 6 mm are set at a spacing of 300 mm. The top 300 mm range of the specimen is the holding region of loading device, in order to prevent local concrete breakage caused by stress concentration. Additional steel bars with a diameter of 6 mm are also set at the junction of the holding region and the research region, with a spacing of 300 mm. Steel plates with a thickness of 14 mm are embedded at the side and top loading points of all specimens for the uniform transfer of horizontal and vertical loads, respectively, as shown in Fig 3(b).

Finally, concrete layers(provided by Shijiazhuang Shuntian Building Materials Co. LTD) on both sides of EPS insulation layer are cast with three different thicknesses of concrete layers: 65 mm, 75 mm and 85 mm. Typically, there are three cast-in-situ methods adopted in practice, such as manually plastered concrete, sprayed concrete and cast-in-place concrete with supporting formwork. The cast-in-situ concrete with supporting formwork is adopted in this experiment, as shown in Fig 3(c).

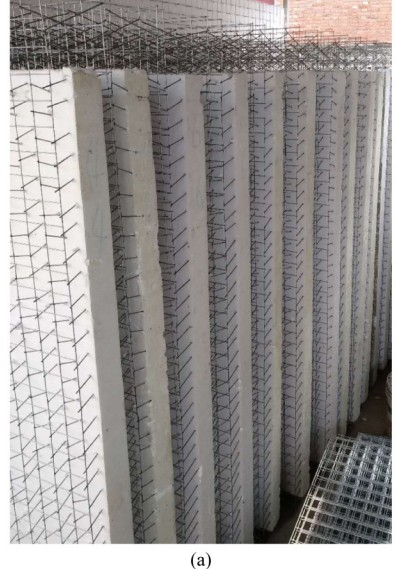 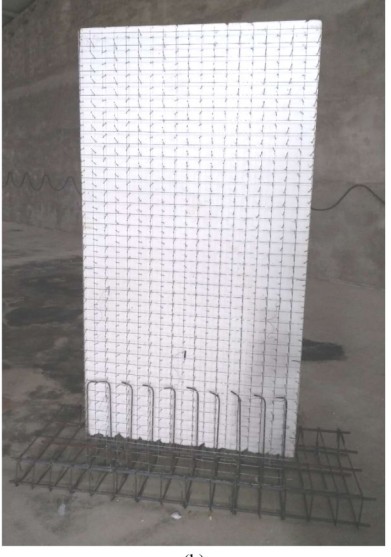 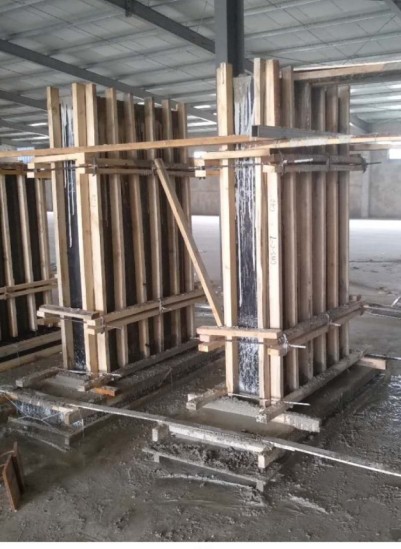

(a)                    (b)                    (c)

**Fig 3. The fabrication process of the specimen.** (a) Steel meshes welding; (b) Reinforcements tying; (c) Concrete casting.

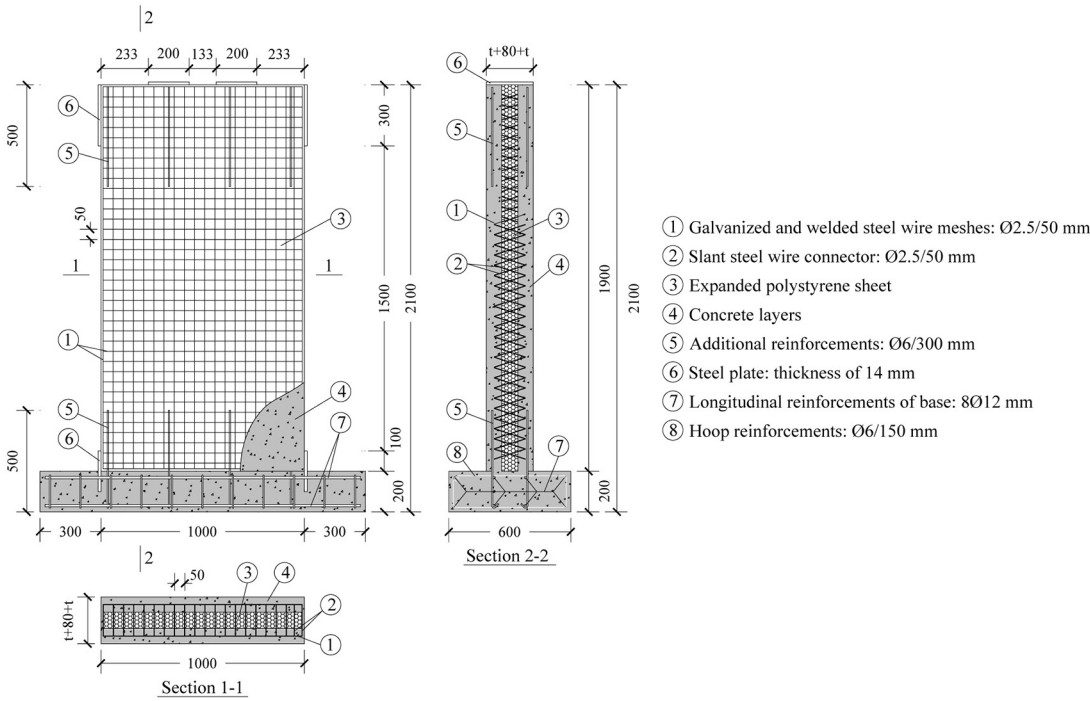

**Fig 4. Layout of reinforcement (unit: mm).**

## Material properties

The materials used in the specimens are typically used in construction sites. Three kinds of concrete (C20, C30 and C40, according to GB/T 50081–2002 Chinese standard for test method of mechanical properties on ordinary concrete) were tested for axial compression strength and splitting tensile strength, as shown in Fig 5(a) and 5(b). The steel wires in the specimen are made of low-carbon galvanized steel wire with a diameter of 2.5 mm, and the additional reinforcements are made of plain steel bars (HPB300, according to GB 50010–2010 Chinese code for design of concrete structures) with a diameter of 6 mm. Tensile tests were carried out respectively, as shown in Fig 5(c) and 5(d). The results of material property tests indicate that the galvanized steel wires used have no obvious yield stage, so the strength when the plastic elongation is 0.2% is taken as the conditional yield strength. The test results are shown in Table 2.

## Test set-up

The test device consists of a reaction frame, a hydraulic actuator, and a sliding device. The holding device is installed on the top of the specimen. The horizontal cyclic load is applied by a double-acting hydraulic actuator through the holding device. The vertical load is applied by the vertical actuator through a load-bearing steel beam on the top of the specimen. The near end of the vertical actuator is a spherical hinge, and the root is fixed on the reaction frame by sliding device. The vertical actuator can slide freely along the loading direction of the horizontal actuator. The specimen foundation beam is fixed on the reaction frame with a high-strength bolt. In order to prevent rigid body slip-off of the specimen along the loading direction, a hydraulic jack is set at the southern end of the foundation beam along the horizontal loading direction to tighten the foundation, as shown in Fig 6. The above equipments are manufactured and supplied by Beijing Baoheyuan Photoelectric Equipment Co. LTD.

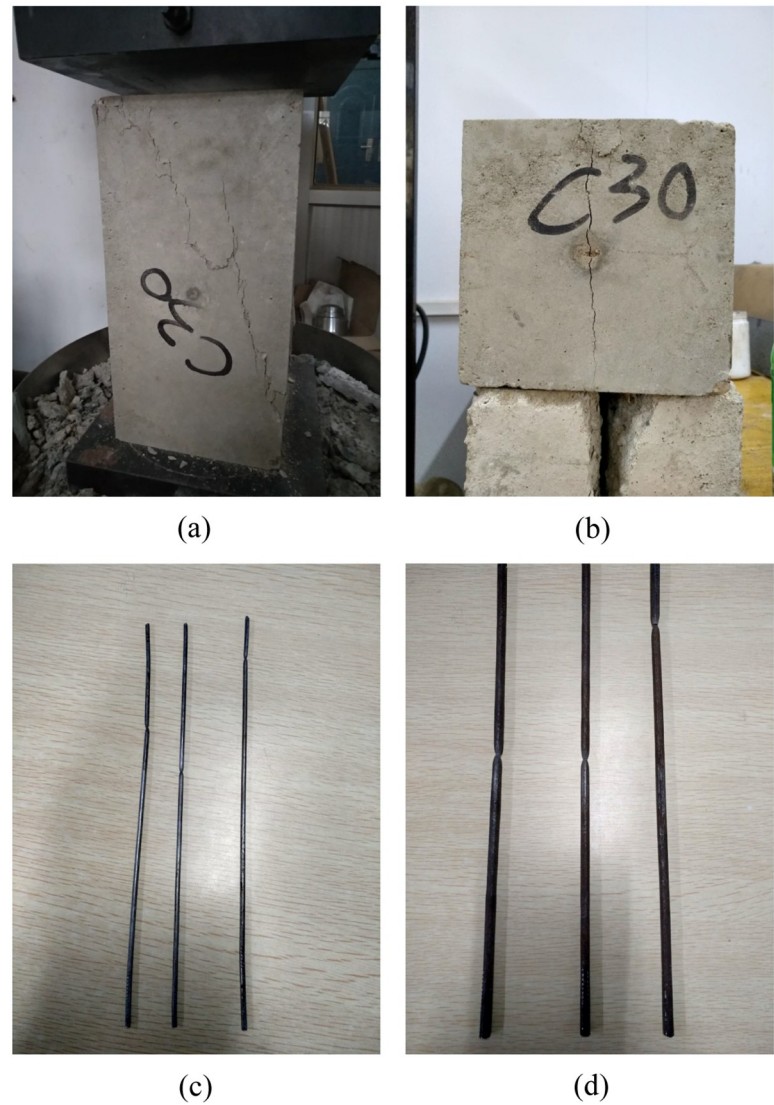

**Fig 5. Appearance of materials after strength tests.** (a) Axial compressive strength test on concrete; (b) Splitting tensile strength test on concrete; (c) Tensile test on low carbon galvanized steel wire; (d) Tensile test on additional reinforcement.

Four digital indicators each with a range of 50 mm and one draw-wire displacement sensor (both are manufactured by Beijing Jinghaiquan Sensor Technology Co. LTD) are applied to each specimen to measure its displacement, as shown in Fig 6:

- Absolute horizontal displacement is measured by digital indicator No. 1 and No. 3 and the draw-wire displacement sensor.

- The horizontal rigid body displacement of the foundation beam is measured by digital indicator No. 2 and No. 4.

During the loading process, the vertical load, horizontal load, and horizontal displacement are automatically collected by the DONGHUA-DH3821 data collection system, while the wall cracking and damage phenomena are observed by the naked eye.

**Table 2. Summary of material properties.**

| Property | | Notation | Average/ MPa | Test standard |
|---|---|---|---|---|
| Axial compressive strength of concrete | C20 | $f_{cp}$ | 24.39 | GB/T 50081–2002 Chinese standard for test method of mechanical properties on ordinary concrete |
| | C30 | | 34.36 | |
| | C40 | | 44.69 | |
| Splitting tensile strength of concrete | C20 | $f_{ts}$ | 1.8 | |
| | C30 | | 2.4 | |
| | C40 | | 2.7 | |
| Conditional yield strength of low carbon galvanized steel wire | | $f_{sy}^{0.2}$ | 435 | GB/T 228.1–2010 Chinese standard for metallic materials-Tensile testing-Part 1: Method of test at room temperature |
| Tensile strength of low carbon galvanized steel wire | | $f_{su}$ | 549 | |
| Yield strength of additional reinforcement | | $f_y$ | 315 | |
| Tensile strength of additional reinforcement | | $f_u$ | 431 | |

## Phenomena of tests

This section presents the cracking and final failure patterns of the single cast-in-place sandwich concrete panel specimens during loading.

At the initial stage of loading, all the specimens show horizontal bending cracks at the edges of wall, and they gradually develop as diagonal shear cracks with the displacement growth. Each specimen is damaged together with the diagonal shear cracks penetration in the lower part of the wall. Finally, the cracks develop to the bottom of the wall, forming a main crack with a considerable width and length in both directions. The two main cracks intersected in an "X" shape, determining the failure mode of the wall. Fig 7 illustrates the final form of failure of CWS-1.

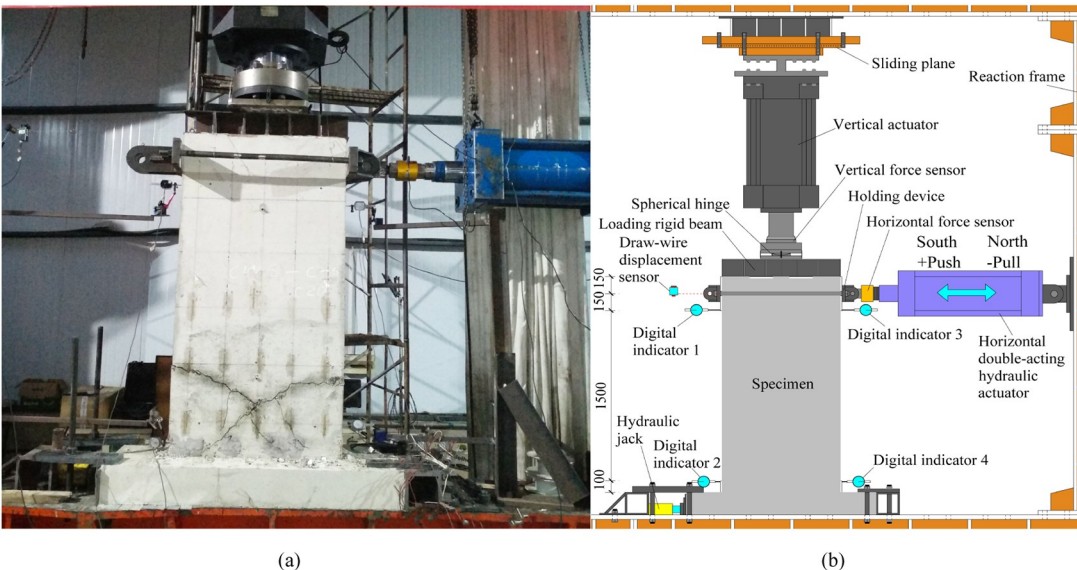

(a)                                                                    (b)

**Fig 6. Test set-up.** (a) Photo of test set-up; (b) Sketch of the test set-up.

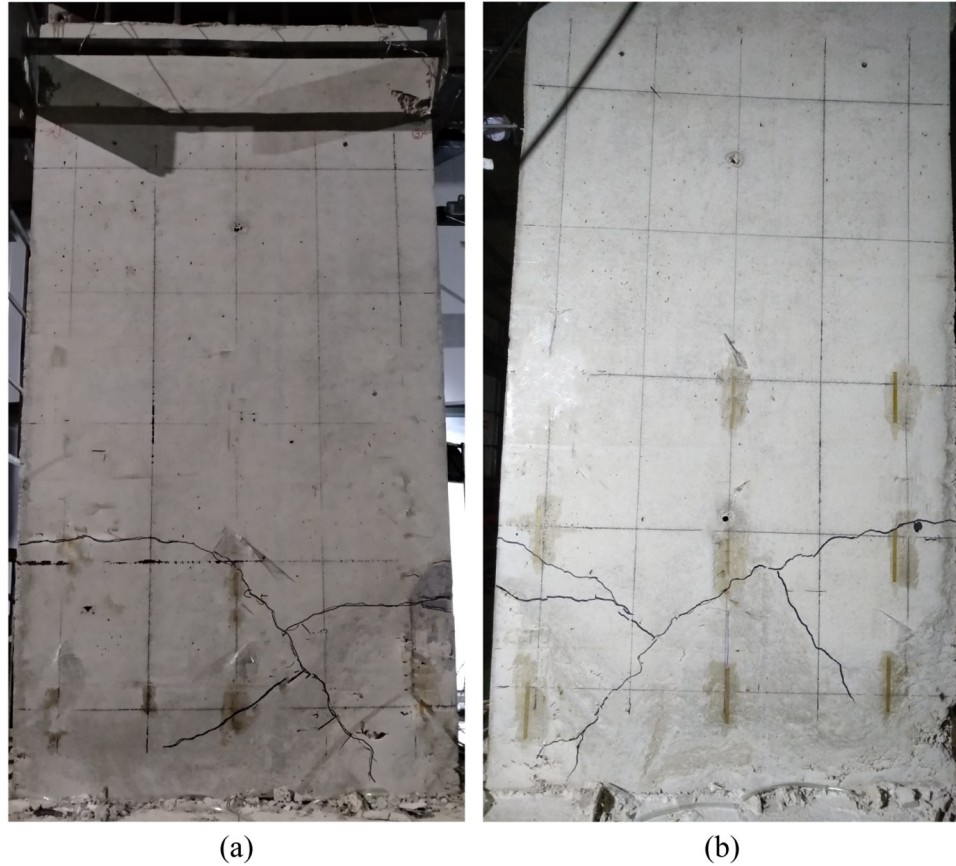

**Fig 7. Final cracking patterns of CWS-1.** (a) East; (b) West.

Observations will be presented in groups according to different research parameters, wherein the panel CWS-1 is a reference specimen.

- For the specimens with the thickness of the concrete layer as the variable: CWS-2 ($t = 65$ mm), CWS-1 ($t = 75$ mm), CWS-3 ($t = 85$ mm), the wire spacing is 50 mm, the reinforcement ratio is 0.13%, the axial compression ratio is 0.06, and the failure modalities of them are similar. Taking the CWS-1 as an example: the axial force applied to CWS-1 is 340 kN, and the ultimate displacement is 0.9% (15 mm). The first crack, nearly horizontal, is a bending crack generated at a height of approximately 630 mm (1/3 of specimen height) above the lower edge of the specimen at the negative cyclic displacement of -0.18% (-3 mm, $F = -114$ kN). When loaded to -0.24% (-4 mm, $F = -132$ kN), the crack extends diagonally as a shear crack. When loaded to -0.72% (-12 mm, $F = -111$ kN), the crack reaches the bottom of the specimen. When the load was increased to +0.5% (+8 mm, $F = +143$ kN) and +0.72% (+12 mm, $F = +136$ kN), a crack is created in the other direction, intersecting the existing crack. It is worth noting that CWS-1 and CWS-2 only produce one main crack in the positive and negative loading direction respectively, and the branch cracks are few; in contrast, the crack distribution of CWS-3 is denser, as shown in Fig 8(a), 8(b) and 8(c).

- For CWS-4 with the wire spacing as the variable, the concrete layer thickness is 75 mm, the vertical axial force applied is 340 kN, the axial compression ratio is 0.06, the wire spacing is 100 mm, the reinforcement ratio is 0.07%, and the ultimate displacement is 1.0% (17 mm).

When the negative cyclic displacement is -0.5% (-8 mm, $F$ = -151 kN), a horizontal crack of 0.1 mm wide is produced at the bottom of the south side of the panel, which is a bending crack. A 0.1 mm wide horizontal crack is generated at the bottom of the north side of the panel when the positive cyclic displacement is +0.7% (+13 mm, $F$ = +189 kN), and the two cracks are connected to each other. When the displacement gradually increases to -1.0% (-17 mm, $F$ = -191 kN), the edge of the panel cracks suddenly with a loud burst and the cracks quickly penetrate to the bottom of the wall. The maximum width of the cracks is 4.5 mm. The cracks start horizontally, and then proceed diagonally. Finally, the specimens are damaged without omen. The reason for this may be the lower reinforcement ratio of CWS-4. In contrast, although the damage of CWS-1 also shows brittleness, it is not sudden, but rather gradual and manageable. See Fig 8 (a) and 8(d).

- For the specimens CWS-5 and CWS-6 with concrete strength as the variable, the concrete layer thickness is 75 mm, the wire spacing is 50 mm, the reinforcement ratio is 0.13%, the vertical axial force applied are 220 kN and 400 kN respectively, the axial compression ratio is 0.06, the ultimate displacements are 1.7% (28 mm) and 1.0% (18 mm), the concrete strength grades of specimens CWS-5 and CWS-6 are C20 and C40 respectively, and the cracking patterns of them are similar. The CWS-5 specimen is taken as an example for description. For CWS-5, a horizontal crack is generated in the center and lower part of the north side of the panel at a positive cyclic displacement of +0.24% (+4 mm, $F$ = +123 kN), which is a bending crack. When the positive cyclic displacement is +0.36% (+6 mm, $F$ = +136 kN), the crack develops diagonally as a shear crack. On the other hand, when the negative cyclic displacement is -0.36% (-6 mm, $F$ = -112 kN), a shear crack is also generated along the diagonal direction, and the two cracks intersect with each other and stretch to the bottom of the specimen. Compared with CWS-1, the failure patterns of the three are similar, but the cracks in CWS-5 are slightly denser, indicating that the strength of concrete has little effect on the failure pattern of single cast-in-situ insulated concrete wall, as shown in Fig 8(a), 8(e) and 8(f).

- For the specimen CWS-7 with the axial compression ratio as the variable, the concrete layer thickness is 75 mm, the wire spacing is 50 mm, the reinforcement ratio is 0.13%, the vertical axial force applied is 600 kN, the axial compression ratio is 0.12, and the ultimate displacement is 1.9% (32 mm). The first crack is observed in the center of the north side of the specimen when the positive cyclic displacement is +0.45% (+8 mm, $F$ = +196 kN). When the negative cyclic displacement is -0.45% (-8 mm, $F$ = -156 kN), another crack is formed in the lower middle part of the south side of the specimen. Both cracks are horizontal. When the displacement reaches 0.8% (14 mm, $F$ = +232 kN), the cracks develop diagonally, and as the displacement gradually increases, cracks continue to occur until the specimen is destroyed. Compared with CWS-1, the crack position of CWS-7 is higher. Although the number of cracks is comparable between the two specimens, CWS-7 has a larger width of cracks, a number of branch cracks, a denser distribution of cracks, more serious damages and spalling. This is seen in Fig 8(a) and 8(g).

## Experimental results and structural parameter study

### Hysteresis curves and skeleton curves

The hysteresis curves and skeleton curves of seven specimens are shown and compared in Figs 9 and 10 respectively. All specimens have similar hysteresis behavior. The specimens exhibit linear elastic behavior with low energy dissipation before cracking. As the horizontal

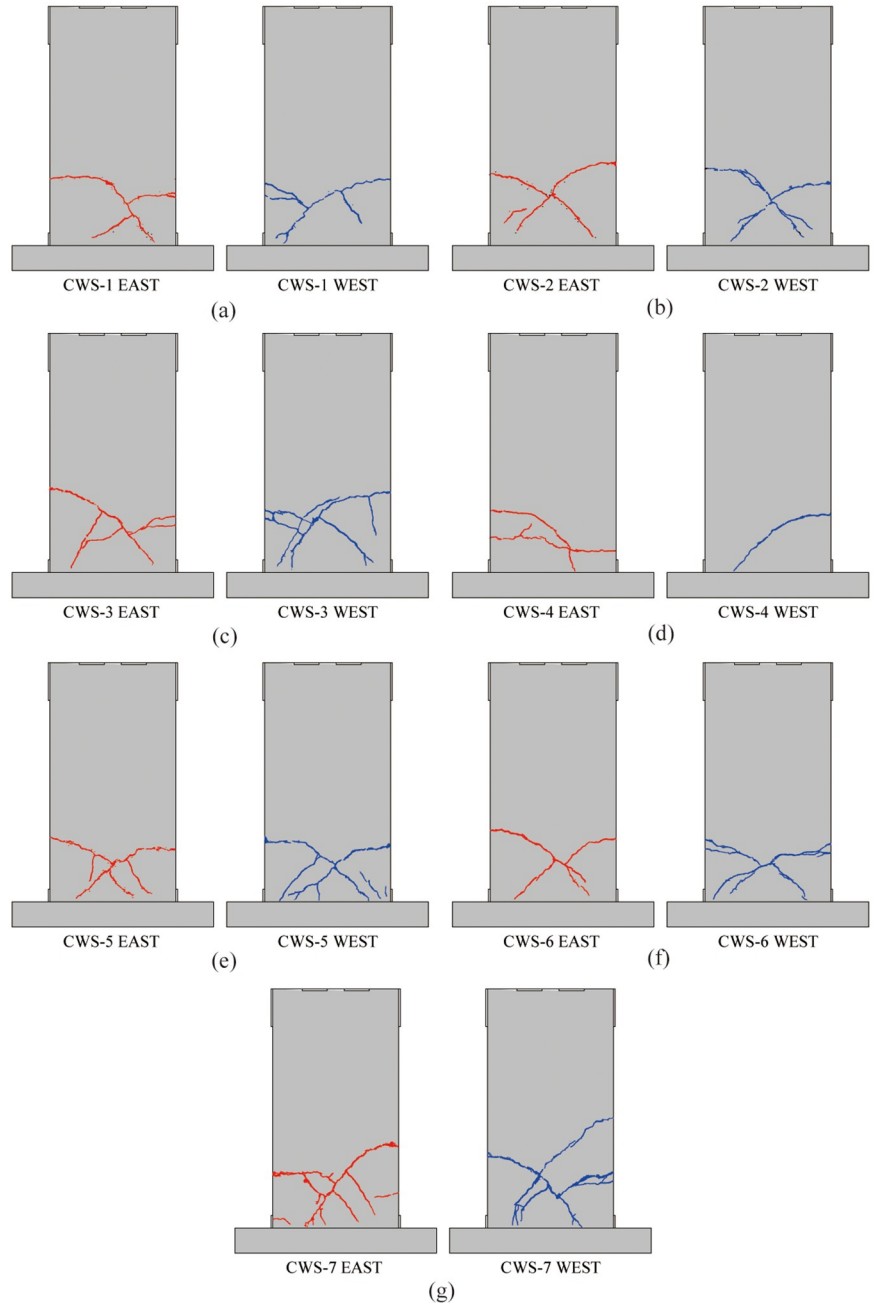

**Fig 8. Final cracking patterns of all specimens.** (a) CWS-1; (b) CWS-2; (c) CWS-3; (d) CWS-4; (e) CWS-5; (f) CWS-6; (g) CWS-7.

displacement gradually increases, hysteresis occurs with high energy dissipation in all specimens.

The hysteresis curves of all specimens are in the same pattern in both positive and negative loading directions. The envelopes of hysteresis curves in the positive and negative directions are asymmetrical, as shown in Fig 9 (the red and blue curves represent the envelope of positive and negative loading, respectively). Except for CWS-2, the envelope in the negative direction

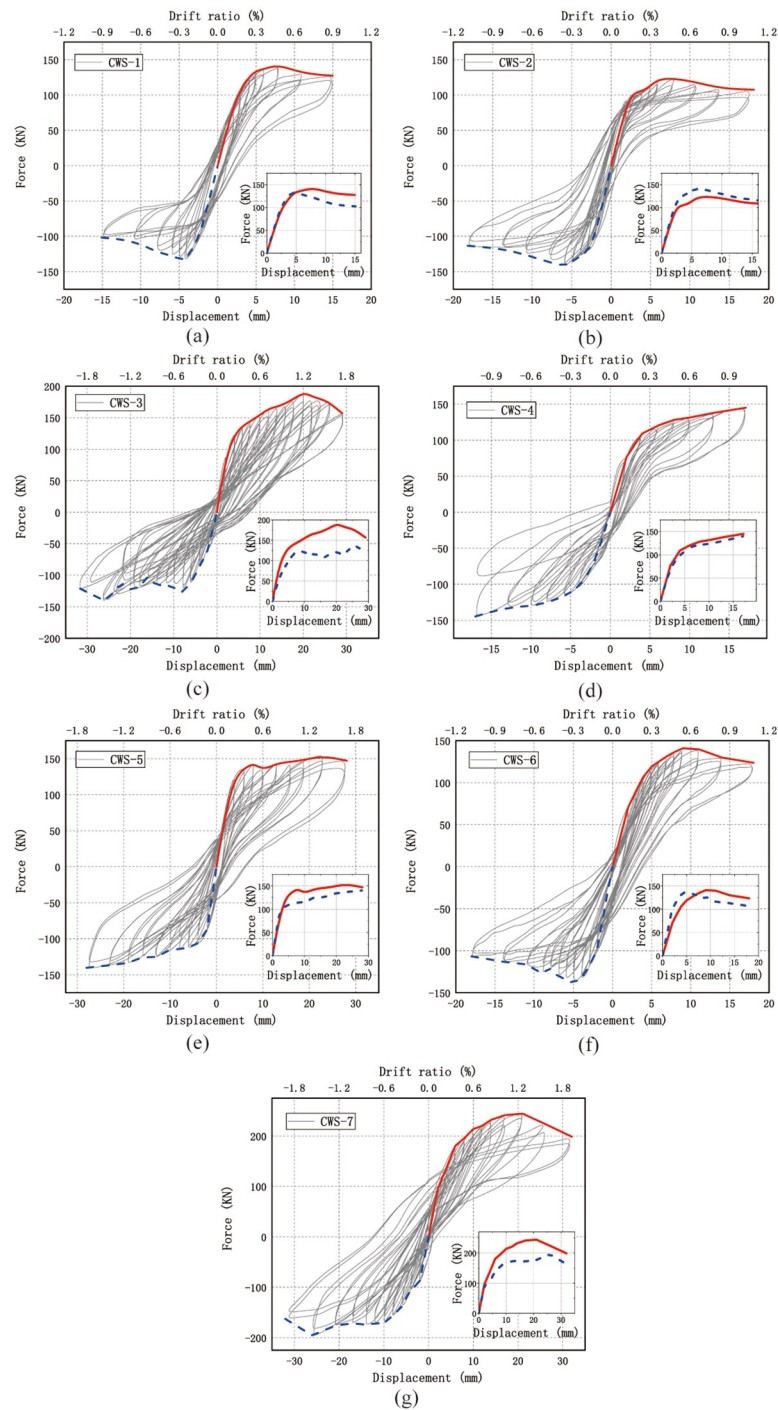

**Fig 9. Final cracking patterns of all specimens.** (a) CWS-1; (b) CWS-2; (c) CWS-3; (d) CWS-4; (e) CWS-5; (f) CWS-6; (g) CWS-7. (Drift ratio refers to the ratio of horizontal displacement to wall height).

is lower than the envelope in the positive direction because the main diagonal crack in the positive direction develops earlier than the crack in the negative direction. The main diagonal crack of CWS-2 occurs in the negative loading direction first, so the envelope in the negative direction is higher. This may be due to fabrication defects.

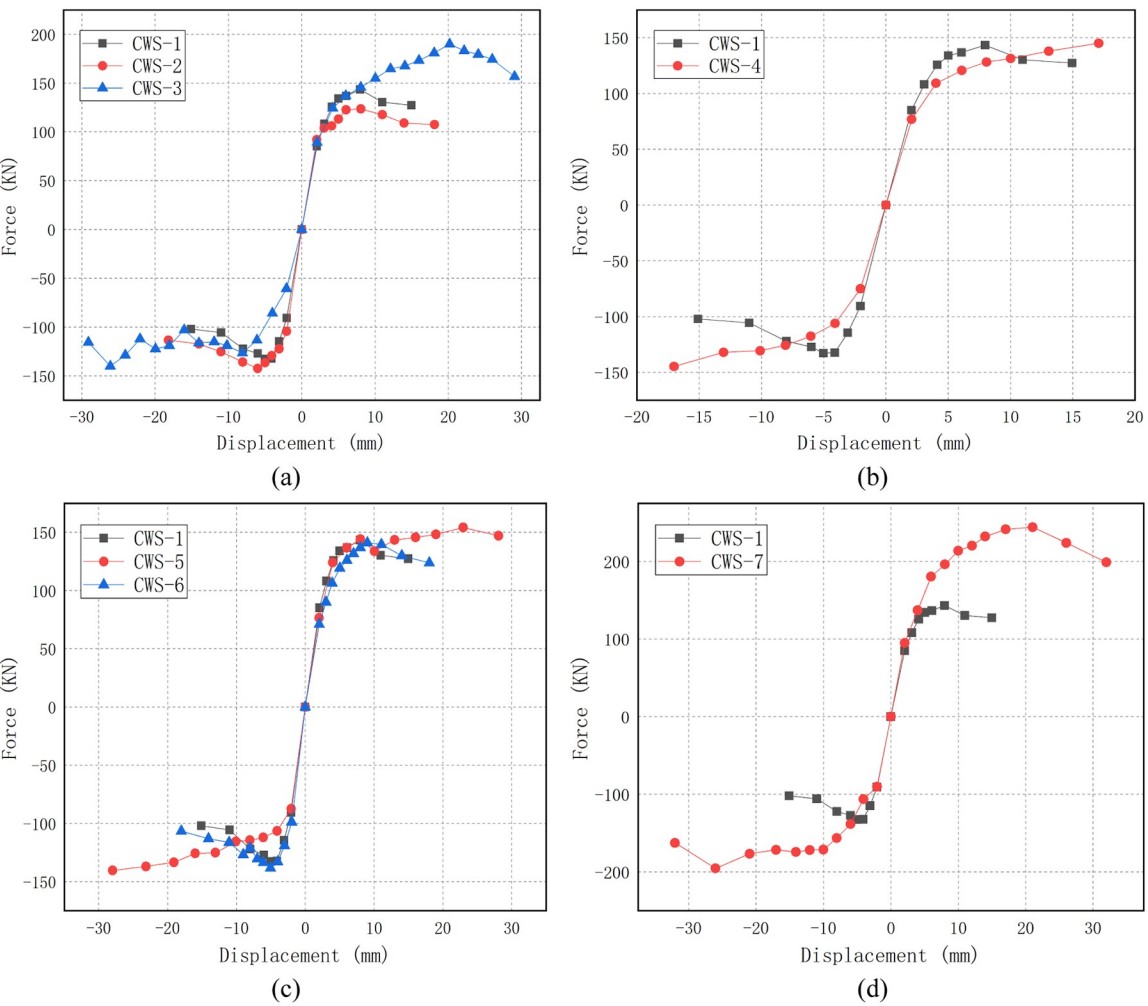

**Fig 10. Skeleton curves.** (a) Walls with different thickness of concrete layers; (b) Walls with different distances of steel wire; (c) Walls with different concrete strength; (d) Walls with different axial compression ratios.

The parameter study of the hysteresis curves is as follows:

- The thicknesses of concrete layers *t* of CWS-1, CWS-2 and CWS-3 are 75 mm, 65 mm and 85 mm, respectively. For CWS-1 and CWS-2, there is no obvious yielding. The hysteresis curves are narrow and pinch phenomenon is obvious. Also, the peak load is reached at the drift ratio of 0.5%, as shown in Fig 9(a) and 9(b). For CWS-3, the hysteresis curve is plump and there is no obvious pinch phenomenon. The peak load of CWS-3 is reached at the drift ratio of 1.2% (See Fig 9(c)). Fig 10(a) shows a comparison of skeleton curves of these three specimens. It is seen that with the increase of the thickness of concrete layer the peak load increases and the residual deformation at unloading increases significantly.

- The spacing s between wires (either horizontal or vertical) in specimens CWS-1 and CWS-4 are 50 mm and 100 mm. There is no descending branch in the hysteresis curve of CWS-4. There is almost no residual deformation and the specimen CWS-4 breaks suddenly. This tells CWS-4 experiences brittle failure. The reinforcement ratio of CWS-4 is low (0.07%). For CWS-4 the steel wire is broken at the same time of the concrete crushing, while for

CSW-1 the strength degradation occurs and structural behavior is more ductile, as shown in Fig 9(a) and 9(d).

- The concrete grades of specimens CWS-1, CWS-5 and CWS-6 are C30, C20 and C40. The hysteresis curves of these three are very similar. There is no obvious yielding and the peak loads are close to each other (peak loads are 138, 147 and 139 kN respectively). The hysteresis curve of CWS-5 is slightly plumper because CWS-5 has a better deformation capacity. The ultimate displacement of CWS-5 is 1.6% comparing with 0.9% of CWS-1 and 1.0% of CWS-6 (See Fig 9(a), 9(e) and 9(f)). In addition, Fig 10(c) shows the skeleton curves of these three specimens are very close to each other. It can be concluded that the horizontal bearing capacity of wall is not significantly affected by concrete strength and that the deformation capacity increases as the concrete strength decreases.

- The axial compression ratios of specimens CWS-1 and CSW-7 are 0.06 and 0.12, respectively, corresponding to axial forces of 340 kN and 600 kN. The other parameters for these two specimens are the same. It is seen in Figs 9(g) and 10(d) that with a bigger ratio of axial compression stress to strength, the bearing capacity is significantly increased and the descending branch of the envelope of the hysteresis curve becomes much steeper. The steeper descending branch means faster strength degradation after the peak load. Note that the residual deformation of CWS-7 in the negative loading direction is significantly larger than that in the positive direction. This is because CWS-7 only develops one main diagonal crack in the positive direction which extends to the bottom of the wall and penetrates through the whole thickness of the wall at a drift ratio of 0.5% (8 mm, $F = +196$ kN). This major crack causes severe pinching effect in the hysteresis curve of CWS-7 in positive direction, while more cracks are generated in the negative direction as the increase of displacement, as shown in Fig 8(g).

### Bearing capacity and ductility

The most common methods to determine the yielding point are the energy equivalent method, the geometric mapping method and the R. Park method [25]. The R. Park method is used here. Fig 11 shows the principle of the R. Park method. In general, 85% of the peak load $F_p$ is taken as the ultimate load, and the corresponding displacement is defined as the ultimate displacement. However, for some specimens (e.g. CSW-5), the failure occurs before the load drops to 85% of the peak load. In this case, the last stage of the cyclic load is taken as the limit state. The load and displacement at the limit state are taken as the ultimate load and ultimate displacement for these specimens.

The displacement ductility factor $\mu$ is defined as the ratio of the ultimate displacement $\Delta_u$ to the yield displacement $\Delta_y$ [26]. The yield load $F_y$, peak load $F_p$, ultimate load $F_u$, the corresponding displacements and displacement ductility factor $\mu$ of all specimens are listed in Table 3.

From Table 3, several conclusions are drawn as follows.

- The concrete layer thickness t has an evident correlation with the horizontal bearing capacity. The horizontal peak load increases by 25% (from 132 to 165 kN) when the concrete layer thickness is increased by 30% (from 65 to 85 mm). In addition, the displacement ductility factor ratios of CWS-1, CWS-2 and CWS-3 (1.0, 1.54 and 0.84) indicate that the specimen has a better deformation ability and ductility with less concrete layer thickness.

- Steel wire spacing s has little influence on the bearing capacity, but significant influence on the ductility of the specimens. The displacement ductility factor μ significantly decreases

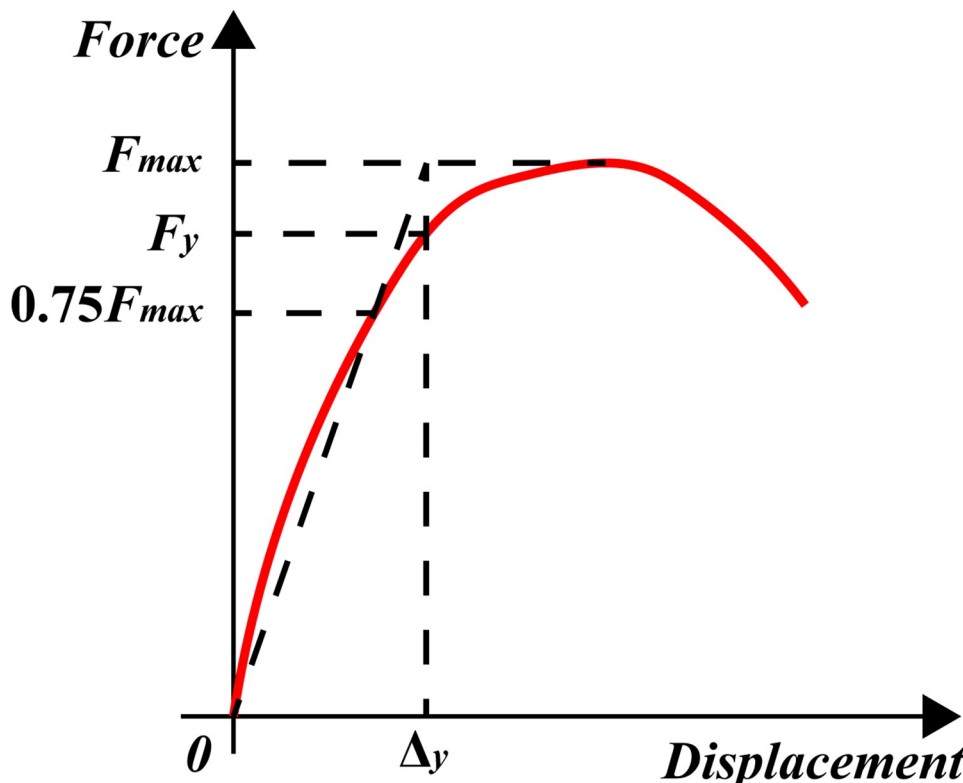

**Fig 11. The yield point determined with R. Park method.**

(from 1.0 to 0.75) when the steel wire spacing is increased (from 50 mm to 100 mm). The smaller the wire spacing is, the greater the ductility is.

- The concrete strength $f_c$ has little effect on the lateral bearing capacity. This is due to a low reinforcement ratio which can't make the full use of concrete strength. The crack develops in the specimens at a low load level, and the lateral bearing capacity is controlled by concrete tensile strength, which fails to give full play to the good compressive strength of concrete. Based on the displacement ductility factor ratios of CWS-1, CWS-5 and CWS-6 (1.0, 1.34 and 1.16), the lower the concrete strength, the better the ductility.

- The axial compression ratio has a significant effect on the lateral bearing capacity. Greater axial compression ratio gives higher bearing capacity. As the axial compression ratio increases from 0.06 to 0.12, the bearing capacity is increased by 58% (from 138 to 219 kN). Meanwhile, the ductility decreases from 1.0 to 0.88.

## Stiffness

The stiffness of a specimen is expressed by secant stiffness $K_i$, which is defined by Eq 1 as in [26]:

$$K_i = \frac{|+F_i| + |-F_i|}{|+\Delta_i| + |-\Delta_i|} \tag{1}$$

where $F_i$ is the peak load at the i-th cycle; i is the corresponding displacement, '+' and '-' signs indicate the direction of loading.

**Table 3. Summary of test results of specimens.**

| Specimen | Loading direction | Yielding | | Peak | | Ultimate | | Ductility | |
|---|---|---|---|---|---|---|---|---|---|
| | | $F_y$ / kN | $\Delta_y$/ mm | $F_p$ / kN | $\Delta_p$/ mm | $F_u$ / kN | $\Delta_u$/ mm | $\mu$ | Ratio |
| CWS-1 | Positive | 125 | 4.07 | 143 | 7.97 | 127 | 15.00 | 3.68 | |
| | Negative | 117 | 3.24 | 133 | 5.03 | 102 | 15.10 | 4.66 | |
| | Average | 121 | 3.65 | 138 | 6.50 | 114 | 15.05 | 4.12 | 1.0 |
| CWS-2 | Positive | 101 | 2.80 | 123 | 8.06 | 107 | 18.11 | 6.46 | |
| | Negative | 120 | 2.90 | 142 | 6.01 | 113 | 18.18 | 6.29 | |
| | Average | 110 | 2.85 | 132 | 7.03 | 110 | 18.14 | 6.37 | 1.54 |
| CWS-3 | Positive | 154 | 9.77 | 190 | 20.18 | 156 | 29.05 | 2.97 | |
| | Negative | 121 | 7.29 | 140 | 26.13 | 115 | 29.10 | 3.99 | |
| | Average | 137 | 8.53 | 165 | 23.15 | 136 | 29.07 | 3.48 | 0.84 |
| CWS-4 | Positive | 116 | 5.34 | —[a] | | 145 | 17.10 | 3.21 | |
| | Negative | 115 | 5.66 | | | 143 | 17.02 | 3.00 | |
| | Average | 115 | 5.50 | | | 144 | 17.06 | 3.10 | 0.75 |
| CWS-5 | Positive | 129 | 4.87 | 154 | 22.95 | 147 | 28.09 | 5.76 | |
| | Negative | 109 | 5.26 | 140 | 28.00 | 140 | 28.00 | 5.32 | |
| | Average | 119 | 5.06 | 147 | 25.47 | 143 | 28.04 | 5.54 | 1.34 |
| CWS-6 | Positive | 120 | 5.16 | 140 | 9.03 | 123 | 18.03 | 3.49 | |
| | Negative | 118 | 2.94 | 138 | 5.04 | 106 | 18.00 | 6.12 | |
| | Average | 119 | 4.05 | 139 | 7.03 | 115 | 18.01 | 4.80 | 1.16 |
| CWS-7 | Positive | 199 | 8.39 | 244 | 21.04 | 198 | 32.00 | 3.81 | |
| | Negative | 165 | 9.21 | 195 | 26.01 | 162 | 32.03 | 3.47 | |
| | Average | 182 | 8.80 | 219 | 23.52 | 180 | 32.01 | 3.64 | 0.88 |

[a] Because of the sudden failure of CWS-4, its skeleton curve has no descending branch, and the maximum load value is taken as the ultimate load value.

Fig 12 shows the secant stiffness versus lateral horizontal displacement of all specimens. It can be observed that the secant stiffness of all specimens decreases rapidly at the initial loading stage, and then tends to be flat until the end of the test.

Under in-plane lateral cyclic load, the stiffness degradation of the wall depends on the degree of damage during the cyclic response. Fig 13 shows the stiffness degradation of all specimens according to the grouping of research parameters, adopting the stiffness degradation ratio (i.e., the ratio of secant stiffness $K_i$ of the i-th cycle to the secant stiffness $K_1$ of the first cycle).

- For CWS-1($t$ = 75 mm), CWS-2($t$ = 65 mm) and CWS-3 ($t$ = 85 mm) (Fig 13(a)), CWS-2 exhibits faster stiffness degradation because the concrete layer of CWS-2 is thinner and the cracks run through the entire concrete layer after the concrete cracks, leading to greater damage to the specimen. In the initial stage of loading, the stiffness degradation rate of CWS-1 and CWS-3 is similar, the concrete layer of CWS-1 cracks when the displacement reaches 4 mm, and the stiffness of CWS-1 decreases rapidly until it is failure. In contrast, the degradation of stiffness of CWS-3 occurred gradually; the reason for this is that the concrete layers of the CWS-3 are thicker and the cracks are developed gradually.

- For CWS-1($s$ = 50 mm) and CWS-4 ($s$ = 100 mm) (Fig 13(b)), at the initial stage of loading (the displacement is less than 6 mm), the stiffness degradation rate is similar, but after that, the stiffness degradation rate of CWS-1 is faster; this may be because in the later stage of

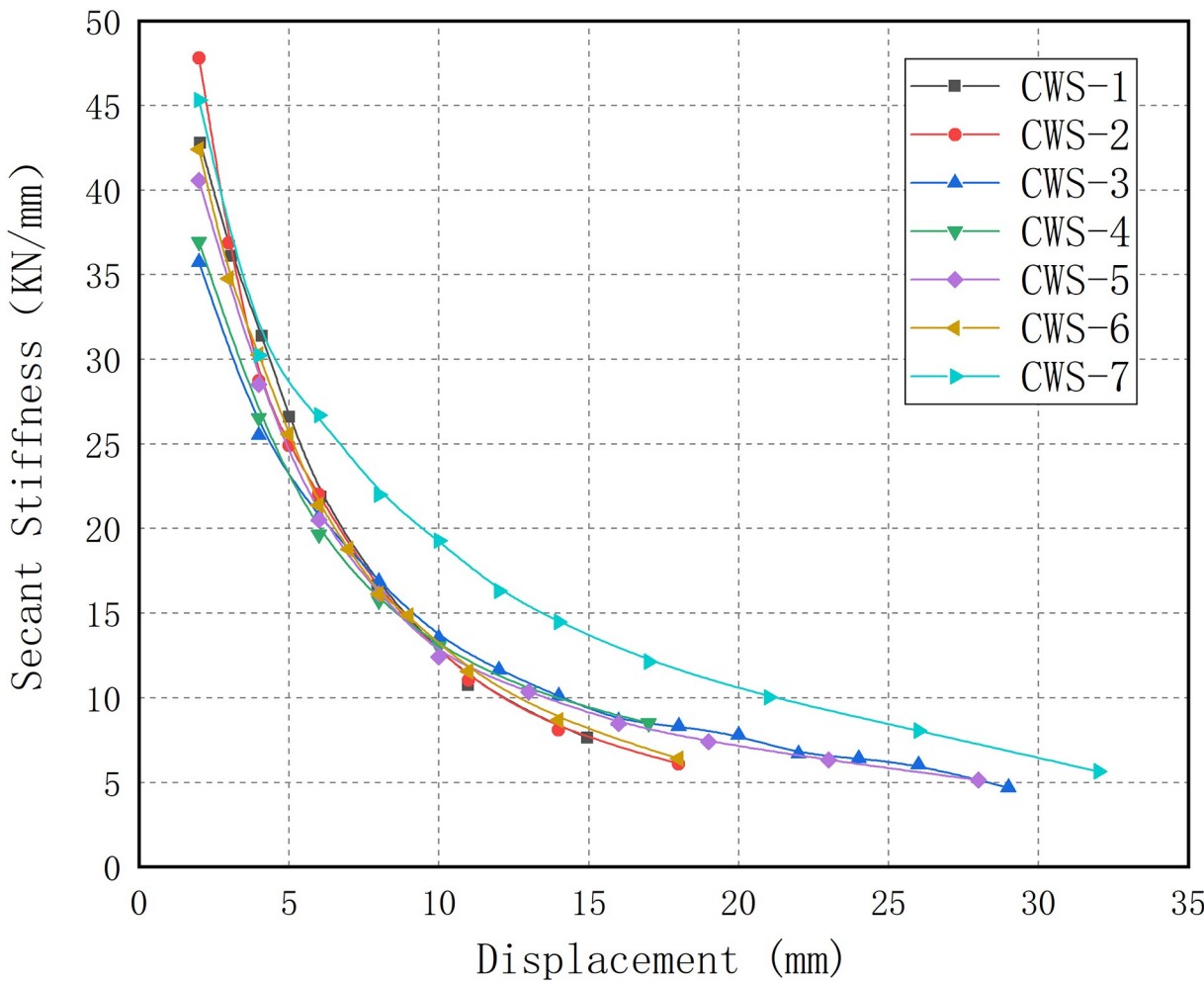

**Fig 12. The secant stiffness versus lateral horizontal displacement.**

loading, no obvious damages appeared to CWS-4 until the displacement reached 17 mm, when it is suddenly destroyed.

- For CWS-1 ($f_{cp}$ = 34.36 MPa), CWS-5($f_{cp}$ = 24.39 MPa) and CWS-6 ($f_{cp}$ = 44.69 MPa) (Fig 13(c)), at the early stage of loading (where the displacement is less than 10 mm), the stiffness degradation curves are almost identical; at the later stage of loading, the stiffness degradation rate of CWS-5 is relatively gentle because the fracture distribution is relatively dense, and the damage of CWS-5 is gradually generated.

- For CWS-1 ($\lambda$ = 0.06) and CWS-7 ($\lambda$ = 0.12) (Fig 13(d)), at the initial stage of loading (where the displacement is less than 4 mm), the stiffness degradation rates are similar; the main diagonal crack of the CWS-1 has been developed to the bottom of the wall after that, causing large damage and rapid stiffness degradation. However, the main diagonal cracks of CWS-7 are not completely connected but gradually develop with the increase of displacement, due to the large vertical axial force. Compared with CWS-1, the damage of CWS-7 is relatively small, so the degradation rate of stiffness is relatively gentle.

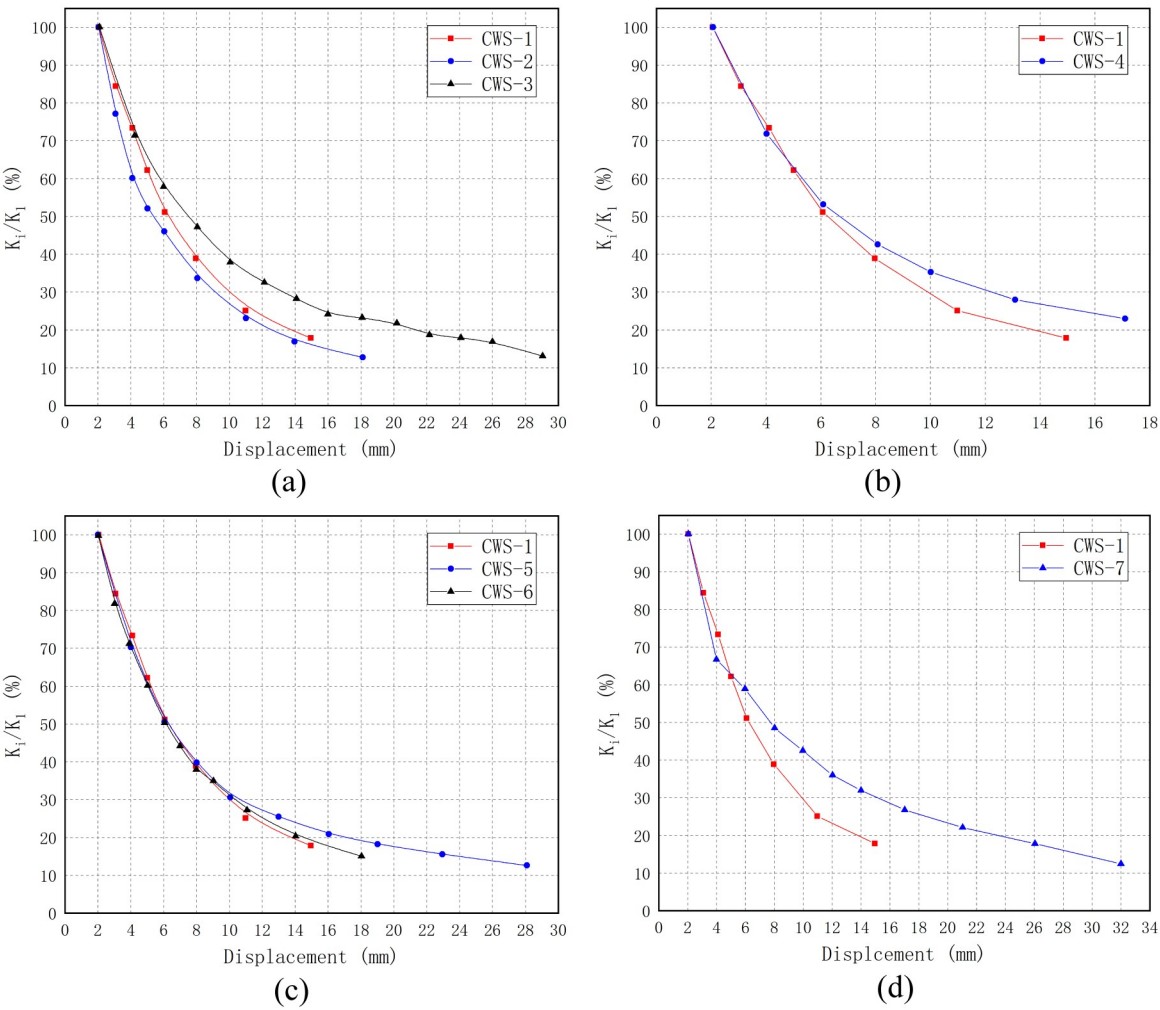

**Fig 13. The stiffness degradation ratio versus loading displacement.** (a) Walls with different thickness of concrete layers; (b) Walls with different distance of steel wire; (c) Walls with different concrete strength; (d) Walls with different axial compression ratio.

### Energy dissipation capacity

Energy dissipation ratio $E$ is often used to measure the energy dissipation capacity of the structure. It is defined by Eq (2), as in [26]:

$$E = \frac{S_{(ABC+CDA)}}{S_{(OBE+ODF)}} \tag{2}$$

where $S_{(ABC+CDA)}$ is the enclosed area by hysteretic loops (the shaded part), $S_{(OBE+ODF)}$ is the total area of two triangles bounded by dotted lines (Fig 14).

The larger the energy dissipation ratio is, the plumper the hysteresis curve is and the stronger the energy dissipation capacity of the structure is. Fig 15 shows the energy dissipation ratio versus the loading displacement of each specimen by grouping the research parameters. It can be seen from Fig 15 that, in the initial stage of loading (before the displacement is about

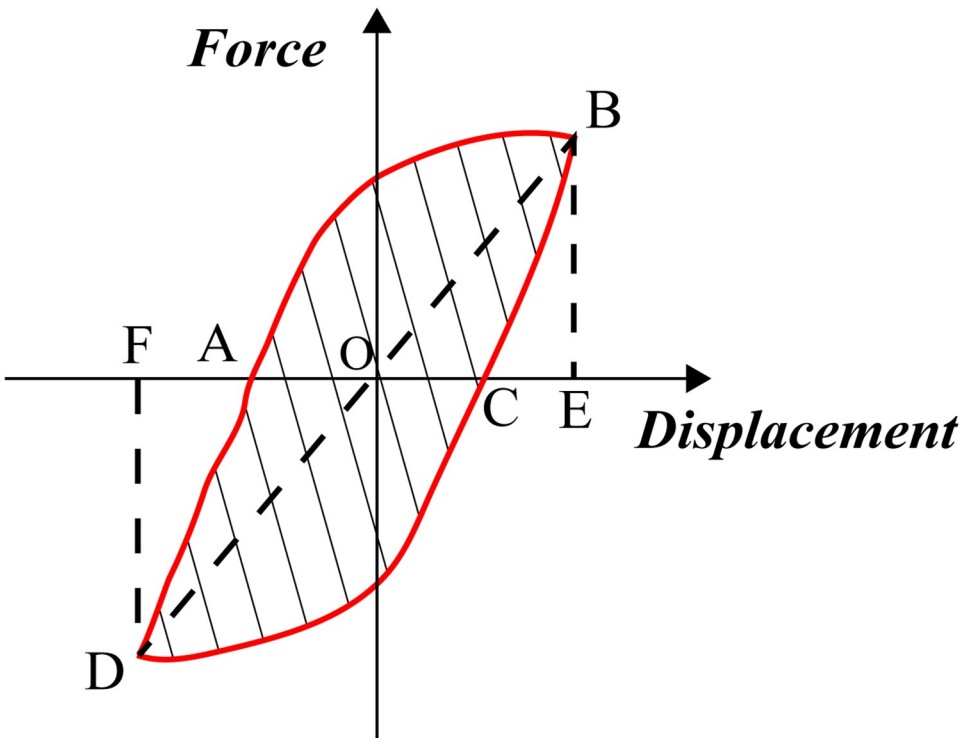

**Fig 14. Schematic diagram of energy dissipation ratio.**

5 mm), the energy dissipation ratio of all specimens decreases rapidly; after that, a continuous trough appeared, and the length and value are different with the different structures or loading devices; finally, the energy dissipation ratio of all specimens shows an upward trend. The reasons for this phenomenon are:

- In the initial stage of loading, the deformation of the specimen is in the elastic stage, and the hysteresis curve can hardly show hysteresis. The hysteresis loop is narrow and long, and the energy dissipation property is low, so the energy dissipation ratio decreases rapidly.

- Since the first crack has appeared, the specimen begins to dissipate energy through the opening and closing of concrete cracks. With the continuous increase of lateral horizontal displacement, the depth and size of existing cracks gradually become larger and new cracks are generated. The bond slip between steel wire and concrete also gradually contributes to energy dissipation. Therefore, the energy dissipation ratio E in Fig 15 experiences a fluctuation range and then begins to rise.

- At the later stage of loading, the main diagonal cracks have formed and gradually spread through the entire section. After the peak point, the hysteresis of the hysteresis curve increases significantly. At this time, the deformation and fracture of the steel wire and friction between cracks of concrete become the main contributors to energy dissipation. Although the energy dissipation ratio starts to increase, due to the low reinforcement ratio and the initial cracks of the specimens, the bearing capacity begins to decline with the increase of displacement, and the test is finished as the specimen approaches the ultimate state.

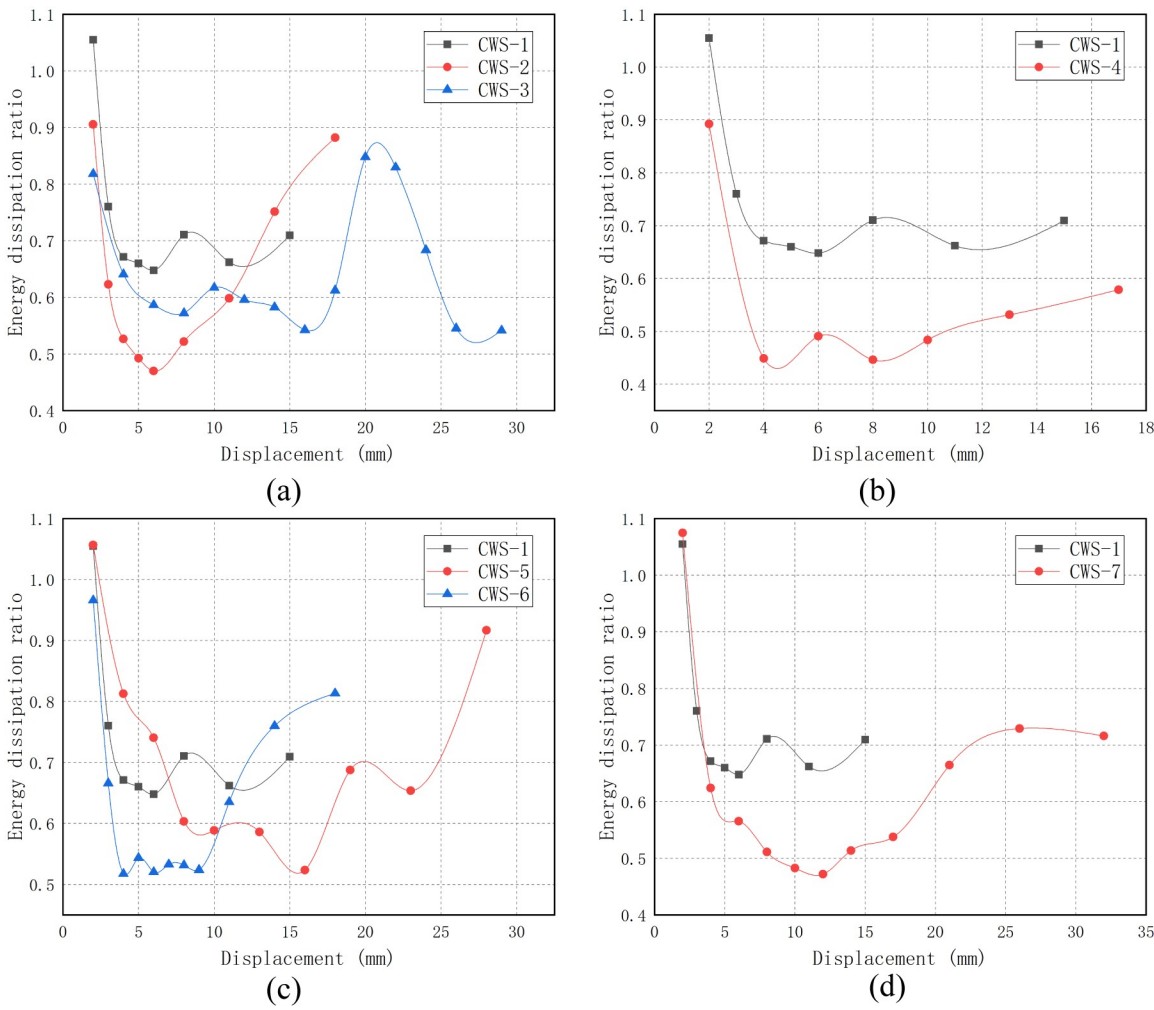

**Fig 15. The energy dissipation ratio versus the loading displacement.** (a) Walls with different thickness of concrete layers; (b) Walls with different distance of steel wire; (c) Walls with different concrete strength; (d) Walls with different axial compression ratio.

From the comparative analysis in Fig 15, the following observations are made:

- The thicker the concrete layer, the better the energy dissipation performance. As shown in Fig 15(a), the energy dissipation ratio of CWS-2 decreases the fastest at the initial stage of loading, because the crack penetrates through the wall earlier.

- The steel wire spacing has a great influence on the energy dissipation of the structure. Because the horizontal and vertical steel wire spacing are equal, the size of the steel wire spacing represents the horizontal and vertical reinforcement ratio of the wall. It can be seen from Fig 15(b) that when the steel wire spacing is low (that is, when the reinforcement ratio is high), the energy dissipation performance is better.

- As seen from Fig 15(c), when the strength of concrete is small, the specimen has a better energy dissipation performance. This is because the higher the strength of concrete is, the worse its ductility is.

- According to Fig 15(d), when the axial compression ratio $\lambda$ is relatively small, the energy dissipation performance is better, which may be due to the fact that when $\lambda$ is relatively large, the specimens are destroyed more violently, and the ductility is poor.

## Conclusions

Pseudo-static tests were carried out on seven full-scale single insulated sandwich concrete cast-in-situ wall specimens. Four different parameters, such as concrete layer thickness, steel wire spacing, concrete strength and axial compression ratio, were considered. The main results obtained from the seven tests can be summarized as follows:

- Under in-plane horizontal cyclic load, all specimens show similar failure modes: in both positive and negative loading directions, a main diagonal crack is found respectively at a height of approximately 630 mm (1/3 of specimen height) above the lower edge of the specimen, and the two main cracks intersected in an 'X' shape. These two cracks predominated and defined the final failure pattern of the wall.

- When all the specimens failed, the concrete layers on either side are not detached from the polystyrene layer in between, which indicates that the slant steel wire shear connector can guarantee the overall working performance of the insulated sandwich concrete cast-in-situ wall.

- The hysteretic curve of the specimens with larger concrete layer thickness is plumper, showing better energy dissipation capacities. The hysteretic curves of specimen with large steel wire spacing show serious pinching phenomenon, small plastic deformation, sudden destruction and poor energy dissipation performance.

- The horizontal bearing capacity is significantly affected by concrete layer thickness and axial compression ratio, but less affected by concrete strength. In addition, a better ductility can be achieved by lowering concrete strength, thickness concrete layer or axial compression ratio.

- When secant stiffness is used as an indicator of stiffness degradation, it is found that the rate of stiffness degradation is gentler when the concrete layer is thicker. It is also found that the strength of concrete has little effect on the rate of stiffness degradation.

- Energy dissipation ratio is used to represent energy dissipation performance. The results show that the concrete layer is thicker, the energy dissipation performance is better when the steel wire spacing is smaller, the concrete strength is lower, and the axial compression is smaller.

## Suggestions

According to the research results of this paper, the following suggestions are given to make the insulated sandwich concrete panel structure used more efficiently in actual construction.

- For the concrete layer thickness, 50~60mm is a reasonable range.

- Improving the reinforcement ratio and adding the lateral constraint members such as RC or profile steel columns can effectively improve the seismic performance.

- Prefabricated ISCP is a good way to simplify and accelerate the construction of this structure.

- To improve the non-continuity of insulation layer in traditional structure, the new joint construction is proposed in this paper, which are shown in Fig 2.

- To improving the insulation of steel shear connectors, the hybrid connectors, i.e., covering the surface of steel elements by using the insulating material, have been proposed.

- In non-load-bearing components, it is a good way to improve the thermal efficiency by using foamed concrete layer as two side plates.

## Supporting information

**S1 Data. Experiment data for all specimens.**
(RAR)

## Acknowledgments

This work was supported by the Natural Science Foundation of Hebei Province (No. E2016210052) and Key Funding Project of Science and Technology Research of Higher Education Institution in Hebei Province (No. ZD2018250). Additionally, this research was also conducted with the financial support of Hebei Xi Jiefa Construction Engineering Co., Ltd.

## Author Contributions

**Conceptualization:** Wentao Qiao.

**Data curation:** Xiaoxiang Yin.

**Formal analysis:** Wentao Qiao, Xiaoxiang Yin.

**Funding acquisition:** Wentao Qiao.

**Investigation:** Xiaoxiang Yin.

**Methodology:** Wentao Qiao.

**Project administration:** Wentao Qiao.

**Resources:** Wentao Qiao.

**Supervision:** Wentao Qiao.

**Validation:** Wentao Qiao.

**Visualization:** Xiaoxiang Yin.

**Writing – original draft:** Xiaoxiang Yin.

**Writing – review & editing:** Shengying Zhao, Dong Wang.

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
