## [Decision Letter · Decision Letter 0]

4 Sep 2019

PONE-D-19-21558

Cyclic loading test study on a new cast-in-situ insulated sandwich concrete wall

PLOS ONE

Dear Dr. Qiao,

Thank you for submitting your manuscript to PLOS ONE. After careful consideration, we feel that it has merit but does not fully meet PLOS ONE’s publication criteria as it currently stands. Therefore, we invite you to submit a revised version of the manuscript that addresses the points raised during the review process.

We would appreciate receiving your revised manuscript by Oct 19 2019 11:59PM. To enhance the reproducibility of your results, we recommend that if applicable you deposit your laboratory protocols in protocols.io, where a protocol can be assigned its own identifier (DOI) such that it can be cited independently in the future. For instructions see: http://journals.plos.org/plosone/s/submission-guidelines#loc-laboratory-protocols

We look forward to receiving your revised manuscript.

Kind regards,

Antonio Riveiro Rodríguez, PhD

Academic Editor

PLOS ONE

1. Please ensure that in your methods sections that you have specified the sources of all materials, equipment and instrumentation used in your study, for example manufacturer or supplier names. This is in line with our reproducibility publication criterion, see  https://journals.plos.org/plosone/s/criteria-for-publication#loc-3

Reviewers' comments:

Reviewer's Responses to Questions

**Comments to the Author**

1. Is the manuscript technically sound, and do the data support the conclusions?

Reviewer #1: Partly

Reviewer #2: Yes

2. Has the statistical analysis been performed appropriately and rigorously? 

Reviewer #1: N/A

Reviewer #2: Yes

3. Have the authors made all data underlying the findings in their manuscript fully available?

Reviewer #1: Yes

Reviewer #2: Yes

4. Is the manuscript presented in an intelligible fashion and written in standard English?

Reviewer #1: Yes

Reviewer #2: Yes

5. Review Comments to the Author

Reviewer #1: This study performed cyclic loading test on 7 sandwich concrete walls for better insulation performance. In the first part, the authors explained the method to improve insulation performance and connection joint details. But, in the tests, the authors evaluated the cyclic behavior of single wall. Thus, the paper writing flow needs to be modified for better clearness. Following comments are recommended for better quality of this paper.

1. Introduction is too long. Particularly, reviewing of existing studies needs to be briefly written.

2. The authors introduced connection methods of walls, but the structural performance of only single wall was evaluated. Thus, the chapter “Connecting joints of walls” can cause misunderstanding to the readers.

3. In general, cylinder strength is used to evaluate the structural performance. Please provide the cylinder strength also.

4. In Fig. 6(a), cracks are observed in footing slab, which affects the test results.

5. Cyclic curve shape is completely different from conventional wall test results. Particularly, in unloading part. Maybe this is because of test setup problem (very small sized footing). The authors need to explain such problem and the reason of strange unloading curve, and it is difficult to compare the structural performance based on test error.

6. How to define yield load? From nominal strength or rebar strain?

7. For direct evaluation of test results, nominal strength of flexure and shear should be provided, and those should be compared with the cyclic curve.

8. High-strength concrete improves shear strength, which reduces or delays shear damage. On the other hand, the authors’ test results exhibited the same performance regardless of concrete strength. It needs to be explained more clearly.

9. The authors used this wall system to improve insulation performance. According to test parameters, insulation performance is also affected. Thus, design recommendation addressing both the structural performance and insulation performance needs to be provided.

Reviewer #2: Paper is very interesting. It is in detail in technical parts and easy to read and understand. Information presented in this paper will be very useful for engineers and researchers.

I have just two comments: 1. Title specified that it is new type of panel (expanded polystyrene sheets embedded into two layer of wire meshes) while this type of panels are not new and it was practiced in construction industry for more than 10 years. However, tests presented in this paper are new. If the authors insisted to claim that it is new it should be further explored and justified.

2. As a results and main conclusion, it is expected that the authors give their recommendation to choose the best system for a real construction work.

6. PLOS authors have the option to publish the peer review history of their article (what does this mean?). If published, this will include your full peer review and any attached files.

Reviewer #1: No

Reviewer #2: No

---

## [Author Response · Author response to Decision Letter 0]

22 Oct 2019

Authors' Replies to Reviewer #1:

1. The introduction of this paper has been modified according to your suggestion. See Manuscript or Revised Manuscript with Track Changes.

2. The section "the structure system" of this paper aims to introduce a complete structure system of insulated sandwich concrete wall, which includes wall composition and the construction of connection joints. Considering the limitation of article length, we design the article content reasonably and the main research content of this paper focuses on the structural performance of single wall under cyclic load. In fact, we have carried out cyclic load tests on both the single wall specimens and the wall-wall junction specimens. As for the mechanical properties of the wall-wall junctions under cyclic load, the research conclusions will be published separately in another paper.

3. So far, the shape and size of concrete specimens have not been completely unified in terms of compressive strength. At present, there are two main shapes of concrete specimens:

●Standard specimen of cylinder: 150mm in diameter and 300mm in height, adopted by the United States, Japan, France, Canada, Australia and other countries.

●Standard cube specimen: 150mm in side length, used in China, Britain, Germany and other countries.

According to the value given by UNESCO's Handbook of Reinforced Concrete, the ratio of the compressive strength of the cylinder fcy to the compressive strength of the cube fcu is 0.80. 

According to the A. M. Neville formula,

Where P is the compressive strength; P6 is the compressive strength of cube with 150mm side length; v is the volume of specimen; h is height of specimen; d is maximum transverse size of specimen.

After conversion, the ratio of the compressive strength of the cylinder fcy to the compressive strength of the cube fcu is 0.81.

In this paper, prismatic concrete specimens are adopted in accordance with Chinese specifications. According to Principle and Analysis of Reinforced Concrete, the ratio of prismatic compression strength fpr to cubic compression strength fcu is 0.76, so it can be seen that prismatic compression strength fpr is very close to cylinder compression strength fcy. Therefore, it is reasonable to use prismatic compression strength fpr to evaluate the structural performance.

4. As for figure 6 (a), there may be a misunderstanding. After careful checking, no damage traces were found in the footing slab. This misunderstanding may be due to the electric wires and projections of the digital indicators, the uncleared steel wires used to fix the formwork and projections on the bottom of the specimen. The original photograph is provided below for verification.

5. The reason why the curve is different from the hysteresis curve of the conventional wall is that the construction and material are different from that of the conventional RC wall but not the small sized footing. To be specific, the reinforcement ratio (0.13%) of the specimens is lower than that of the conventional RC wall, and the steel wires have no obvious yield point.

Taking specimen 2 as example, after the specimen cracks, the width and length of the crack increase with the increasing of displacement, and then the stiffness decreases sharply. When the cracks spread to the bottom of the wall, the horizontal displacement applied to the specimen is mainly resisted by cracks’ opening or closing, so the displacement increases significantly but the load increases a little. As the tensile fracture happens to the steel wires gradually, that the cracks in the compression zone contact will generate friction under the action of axial pressure, so the load continues to keep rising. When unloading in the opposite direction, the existing cracks continue to open or close, the friction between cracks and the new cracks still play a role, the bearing capacity no longer increases until the cracks completely penetrate the whole specimen. All of these factors cause the difference of the curve shape.

6. The definition of the yield load has been made in the section "Bearing capacity and ductility" in this paper. It is defined from nominal strength by R. Park method.

7. The description in this paper has been modified according to your suggestion, and the description of strength value has been added to the manuscript. See in Revised Manuscript with Track Changes.

8. The specimens in this paper is not equipped with lateral constraint members, and the reinforcement ratio of the specimens is low, which can’t make the full use of concrete strength. The crack develops in the specimens at a low load level, and the lateral bearing capacity is controlled by concrete tensile strength, which fails to give full play to the good compressive strength of concrete. Therefore, specimens 1, 5 and 6 showed similar performance despite different concrete grades.

This explanation has been added to the section ‘Bearing capacity and ductility’. See Revised Manuscript with Track Changes.

9. In terms of the structural performance, the following suggestions are given:

●For the concrete layer thickness, 50~60mm is a reasonable range.

●Improving the reinforcement ratio and adding the lateral constraint members such as RC or profile steel columns can effectively improve the seismic performance.

●Prefabricated ISCP is a good way to simplify and accelerate the construction of this structure.

About the thermal performance, some suggestions are listed as follows:

●To improve the non-continuity of insulation layer in traditional structure, the new joint construction is proposed in this paper, which are shown in figure 2.

●To improving the insulation of steel shear connectors, the hybrid connectors, i.e., covering the surface of steel elements by using the insulating material have been proposed.

●In non-load-bearing components, it is a good way to improve the thermal efficiency by using foamed concrete layer as two side plates.

These suggestions have been added to the manuscript in section "Suggestion". See Revised Manuscript with Track Changes.

Authors' Replies to Reviewer #2:

Many thanks to the positive comments.

1. ISCP (Insulated Sandwich Concrete Panel) structures have indeed been used in the construction industry for more than a decade. A typical ISCP consists of concrete layers, insulation layer and shear connectors. The concrete layers locate on both sides of the insulation layer, which are connected by shear connectors. However, the traditional ISCP is improved by adding steel tubes and additional reinforcements in this paper. In addition, special constructions of special-shaped columns and additional reinforcements are adopted as well to ensure both the continuity of the insulation layer and strength of wall-wall junction.

2. According to the research results of this paper, the following suggestions are given to make the insulated sandwich concrete panel structure used more efficiently in actual construction.

In terms of the structural performance, the following suggestions are given:

●For the concrete layer thickness, 50~60mm is a reasonable range.

●Improving the reinforcement ratio and adding the lateral constraint members such as RC or profile steel columns can effectively improve the seismic performance.

●Prefabricated ISCP is a good way to simplify and accelerate the construction of this structure.

About the thermal performance, some suggestions are listed as follows:

●To improve the non-continuity of insulation layer in traditional structure, the new joint construction is proposed in this paper, which are shown in figure 2.

●To improving the insulation of steel shear connectors, the hybrid connectors, i.e., covering the surface of steel elements by using the insulating material have been proposed.

●In non-load-bearing components, it is a good way to improve the thermal efficiency by using foamed concrete layer as two side plates.

These suggestions have been added to the manuscript in section "Suggestion". See Revised Manuscript with Track Changes.

---

## [Decision Letter · Decision Letter 1]

29 Oct 2019

Cyclic loading test study on a new cast-in-situ insulated sandwich concrete wall

PONE-D-19-21558R1

Dear Dr. Qiao,

We are pleased to inform you that your manuscript has been judged scientifically suitable for publication and will be formally accepted for publication once it complies with all outstanding technical requirements.

With kind regards,

Antonio Riveiro Rodríguez, PhD

Academic Editor

PLOS ONE

Reviewers' comments:

Reviewer's Responses to Questions

**Comments to the Author**

1. If the authors have adequately addressed your comments raised in a previous round of review and you feel that this manuscript is now acceptable for publication, you may indicate that here to bypass the “Comments to the Author” section, enter your conflict of interest statement in the “Confidential to Editor” section, and submit your "Accept" recommendation.

Reviewer #1: All comments have been addressed

Reviewer #2: All comments have been addressed

2. Is the manuscript technically sound, and do the data support the conclusions?

Reviewer #1: Yes

Reviewer #2: Yes

3. Has the statistical analysis been performed appropriately and rigorously? 

Reviewer #1: N/A

Reviewer #2: Yes

4. Have the authors made all data underlying the findings in their manuscript fully available?

Reviewer #1: Yes

Reviewer #2: Yes

5. Is the manuscript presented in an intelligible fashion and written in standard English?

Reviewer #1: Yes

Reviewer #2: Yes

6. Review Comments to the Author

Reviewer #1: The sandwich concrete walls may be useful in practice. The authors addressed well the reviewer's comments. It is acceptable.

Reviewer #2: (No Response)

7. PLOS authors have the option to publish the peer review history of their article (what does this mean?). If published, this will include your full peer review and any attached files.

Reviewer #1: No

Reviewer #2: No

---

## [Editor Report · Acceptance letter]

15 Nov 2019

PONE-D-19-21558R1 

Cyclic loading test study on a new cast-in-situ insulated sandwich concrete wall 

Dear Dr. Qiao:

I am pleased to inform you that your manuscript has been deemed suitable for publication in PLOS ONE. Congratulations! Your manuscript is now with our production department. 

With kind regards,

on behalf of

Dr. Antonio Riveiro Rodríguez 

Academic Editor

PLOS ONE